# Exploring Blue Spaces’ Effects on Childhood Leukaemia Incidence: A Population-Based Case–Control Study in Spain

**DOI:** 10.3390/ijerph19095232

**Published:** 2022-04-25

**Authors:** Carlos Ojeda Sánchez, Javier García-Pérez, Diana Gómez-Barroso, Alejandro Domínguez-Castillo, Elena Pardo Romaguera, Adela Cañete, Juan A. Ortega-García, Rebeca Ramis

**Affiliations:** 1Albacete University Hospital, 02006 Albacete, Spain; 2Cancer and Environmental Epidemiology Unit, Department of Epidemiology of Chronic Diseases, National Center for Epidemiology, Instituto de Salud Carlos III (Carlos III Institute of Health), 28029 Madrid, Spain; jgarcia@isciii.es (J.G.-P.); a.dominguez@isciii.es (A.D.-C.); rramis@isciii.es (R.R.); 3Centre for Biomedical Research in Epidemiology & Public Health (CIBER Epidemiología y Salud Pública—CIBERESP), 28029 Madrid, Spain; dgomez@isciii.es; 4National Center for Epidemiology, Instituto de Salud Carlos III (Carlos III Institute of Health), 28029 Madrid, Spain; 5Spanish Registry of Childhood Tumours (RETI-SEHOP), University of Valencia, 46010 Valencia, Spain; elena.pardo@uv.es (E.P.R.); adela.canete@uv.es (A.C.); 6Pediatric Environmental Health Speciality Unit, Department of Paediatrics, Environment and Human Health (EH2) Lab., Institute of Biomedical Research, IMIB-Arrixaca, Clinical University Hospital Virgen de la Arrixaca, 30120 Murcia, Spain; ortega@pehsu.org; 7European and Latin American Environment, Survival and Childhood Cancer Network (ENSUCHICA), 30120 Murcia, Spain

**Keywords:** urban blue spaces, environmental factors, childhood cancer, childhood leukaemia, incidence, spatial epidemiology

## Abstract

Background: Blue spaces have been a key part of human evolution, providing resources and helping economies develop. To date, no studies have been carried out to explore how they may be linked to paediatric oncological diseases. Objectives: To explore the possible relationship of residential proximity to natural and urban blue spaces on childhood leukaemia. Methods: A population-based case–control study was conducted in four regions of Spain across the period 2000–2018. A total of 936 incident cases and 5616 controls were included, individually matched by sex, year of birth and place of residence. An exposure proxy with four distances (250 m, 500 m, 750 m, and 1 km) to blue spaces was built using the geographical coordinates of the participants’ home residences. Odds ratios (ORs) and 95% confidence intervals (95%CIs) for blue-space exposure were calculated for overall childhood leukaemia, and the acute lymphoblastic (ALL) and acute myeloblastic leukaemia (AML) subtypes, with adjustment for socio-demographic and environmental covariates. Results: A decrease in overall childhood leukaemia and ALL-subtype incidence was found as we came nearer to children’s places of residence, showing, for the study as a whole, a reduced incidence at 250 m (odds ratio (OR) = 0.77; 95%CI = 0.60–0.97), 500 m (OR = 0.78; 95%CI = 0.65–0.93), 750 m (OR = 0.80; 95%CI = 0.69–0.93), and 1000 m (OR = 0.84; 95%CI = 0.72–0.97). AML model results showed an increasing incidence at closest to subjects’ homes (OR at 250m = 1.06; 95%CI=0.63–1.71). Conclusions: Our results suggest a possible association between lower childhood leukaemia incidence and blue-space proximity. This study is a first approach to blue spaces’ possible effects on childhood leukaemia incidence; consequently, it is necessary to continue studying these spaces—while taking into account more individualised data and other possible environmental risk factors.

## 1. Introduction

In the present moment, more than half of the world’s population lives in cities that are undergoing rapid changes, and natural spaces are becoming more relevant [1]. Blue and green spaces have been considered an important part of urban development plans. Many cities are trying to incorporate parks and/or riverside paths to their urban structure with the aim of enhancing inhabitants’ health [2,3]. 

However, this is not really anything new. Blue spaces have been a key part of human development [4]. They are defined as visible surface water such as coastal water, lakes, and rivers [5]. They provide us with sources of food and drinking water as well as facilitating transport, commerce, and power generation. For these reasons, many of the world’s biggest cities are situated near blue spaces and a great number have created extensive blue urban network infrastructure to take advantage of these spaces’ benefits [6]. On the other hand, pollution of the oceans and rivers is worsening, widespread, and poorly controlled in most countries. This pollution is often heaviest near the coast as a consequence of the intensive human activities.

As the population and urban areas grow rapidly, the number of researchers trying to establish relationships between blue spaces and health has increased. As cities’ urban areas have expanded, blue spaces are generating more interest due to their possible restorative and recreational potential for the surrounding population. According to the available literature, these spaces have been positively associated with improved well-being, mental health, and physical activity promotion [7]. Although these relationships are less well-established than for urban green spaces, it has been hypothesised that blue spaces may follow similar causal pathways to those identified for urban green spaces [8]. These pathways can be organised into three domains which may act together: reducing harm, restoring, and building capacities [9]. Focusing on the infant population, despite the heterogeneity of the studies carried out in terms of design, exposure, outcomes, and measurement tools, many of them have also related blue spaces with positive mental effects and physical activity and reduced obesity [10,11,12]. However, these spaces are related with some direct and indirect negative effects for the population living near them [13,14]. Nevertheless, there is as yet no evidence of possible links between these spaces and infant oncology. 

Leukaemia represents 30% of all childhood cancers, and is the most common type of cancer in children [15]. Although its aetiology remains largely unknown, the distinct incidence patterns by age, sex, or geography distribution may suggest a potential environmental role. In fact, during the last few decades, several studies have explored environmental factors such as benzene exposure, magnetic fields, crop fields, air pollution, ionising radiation, socioeconomic status, industrial and urban sites, and radon [16,17,18,19,20]. A possible association between lower childhood leukaemia incidence and proximity to different forms of urban green space has been reported [21]. In this context, this article is the first attempt to explore the possible link between residential proximity to blue spaces and childhood leukaemia and its subtypes: acute lymphoblastic leukaemia (ALL) and acute myeloblastic leukaemia (AML).

## 2. Materials and Methods

### 2.1. Study Design

A childhood leukaemia case–control population study was conducted in four different Spanish regions: Vizcaya, Barcelona, Valencia, and Murcia, covering the period 2000–2018 (Figure 1). These four regions were selected due to being important urban cores where it was easier to obtain cases for the study with purpose of increase the studio power.

The data used for the study were from children aged 0 to 14 with leukaemia diagnoses between 2000 and 2018. The Spanish Registry of Childhood Tumours (RETI-SEHOP) registered the incident cases. RETI-SEHOP collects information from cases of childhood cancer from hospitals’ paediatric oncology units all over Spain [22]. The average cover of this database is higher than 85% for each of the selected regions. Controls were extracted using random sampling from the Birth Registry of the Spanish Statistical Office (*Instituto Nacional de Estadistica*, INE). They were individually matched to cases in a 6:1 ratio by sex, year-of-birth, and region-of-residence to address possible geocoding errors in the controls chosen. 

All cases’ home addresses at the time of diagnosis were successfully geocoded, and were included in the RETI-SEHOP database. Controls’ coordinates were obtained from the Birth Registry, and were extracted from the mother’s home address as listed on the birth certificate. They were given a 30 m random error to preserve participants’ anonymity. Cases and controls’ residence coordinates were projected into the ETRS89/UTM zone 30N(EPSG:25830) using QGIS software. We excluded controls whose geocoded coordinates were outside the selected regions. 

### 2.2. Blue-Space Selection and Exposure Measurement

We selected blue spaces from the Spanish Land Use Information System (SIOSE) database provided by the Spanish National Geographic Institute website (IGN) [23]. This database divides the terrain into areas (polygons) classifying them in different aggregation levels. The first level describes the land’s use-type, and the second level describes land composition using percentages. SIOSE database counts on 56 first-level use-types and 40 s-level use-types. The minimum mapping unit for urban areas is 1 ha. Cartographic databases from 2005, 2009, and 2014 were used. The dataset from 2005 was used for the participants born between 2000 and 2007, the 2009 dataset for those between 2008 and 2012, and the 2014 dataset for those born between 2013 and 2018.

For this study, the levels and sub-levels that were directly related with natural and urban blue spaces were selected: waterways, marine and continental wetlands, lakes, ponds, reservoirs, coastal lagoons, estuaries, and seas. Artificial blue spaces built for agricultural purposes were not included. Then, buffers for distances of 250 m, 500 m, 750 m, and 1 km from the participants’ home residences were constructed. Subsequently, we designed three exposure groups according to the presence of blue spaces in those buffers: (a) the “reference group” was defined as those subjects who did not have any identifiable blue space in their 1 km exposure buffers, (b) the “exposure group” was established for those children that had blue space at the studied distances, (c) and finally, those children not accounted for in one of the previous groups for each distance were classified as an “intermediate group”. 

### 2.3. Covariates 

#### 2.3.1. Sociodemographic Covariates

We included, as potential confounders, sex, birth year, degree of urbanisation (DGUR), socio-economic status (SES), and activity rate in the analysis. Although SES data were not available at an individual level, data at census-tract level were taken. SES combines information regarding the activity, professional situation, and occupation of the heads of families in each census tract ranging from 0.46 (worst) to 1.57 (best) [24]. In addition, activity rate was included as a confounding variable as well, which was defined as the quotient between adult people working and the population 16 years old or over in the census tract. Both variables were extracted from the 2001 INE Census.

Indicators of urbanisation were assigned using data from the European DGUR data-set [25]. It classifies municipalities into thinly, intermediately, or densely populated areas based on a criterion of geographical contiguity in combination with a minimum population threshold (1 km^2^ population grid cells). As the selected regions have higher-density population compared to most Spanish regions [26], we re-structured DGUR into two groups: densely populated areas and non-densely populated areas.

#### 2.3.2. Environmental Covariates

Applying data available from the European Environment Agency (EEA), particulate matter with a diameter of less than 10 µm (PM_10_) around each child’s home levels was assigned. This agency has produced European interpolated air-quality maps since 2006. These maps are derived primarily from database stations that monitor data across the entire continent. Each grid of the map covers an area of 100 km^2^ [27]. For our study, we obtained PM_10_ levels for the chosen regions for all possible study years and we estimated mean PM_10_ level for each child.

As Markevych et al. and Dzhambov et al. proposed, we estimated surrounding greenness using two different methods [9,28]. On one side, we used Normalized Difference Vegetation Index (NDVI) [29]. We looked for cloud-free Landsat TM images with the highest possible NDVI value (i.e., spring–summer) from each SIOSE database year used (2005, 2009, and 2011). These images were taken from NASA’s Earth Observing System Data and Information System website (EOSDIS) [30]. After image selection, we estimated the amount of photosynthetically active greenness to construct the buffers. For the other approach, a land-use database was used. 

For green space selection, we also used the SIOSE database. To do this, we selected spaces whose level specifications were linked directly with urban green spaces. We followed the same method explained in a previous study to measure the exposure to urban green spaces, applying the same buffer distances as for blue spaces [21]. This exposure measure was categorised into five levels where Quintile 1 (Q1) represents the lowest urban-green-space exposure (reference group).

### 2.4. Statistical Analysis

The odds ratios (ORs) and 95% confidence intervals (95%CIs) associated with blue-space exposure were estimated using fit mixed multiple unconditional logistic regression models that include all the above-mentioned covariates as potential confounders. These all-independent regression models were estimated for overall childhood leukaemia, ALL and AML subtypes, and the sensitivity groups (explained in the following methodology sub).

The statistical programs Microsoft Excel 365^®^ (number version 2203, Madrid, Spain), R^®^ version 4.1.1, STATA^®^ version 15, and the geographic information system QGIS^®^ version 3.18.2 were used. 

### 2.5. Sensitivity Analysis

We decided to perform two different sensitivity analyses. The first was carried out to evaluate a possible misclassification problem due to residential mobility. A matching strategy to find cases with the same address at the time of birth (birth certificate) and diagnosis (included in the RETI-SEHOP registry) was developed. The aim of the second sensitivity analysis was evaluate possible prenatal period exposure. To this end, we decided to select all cases younger than 3 years old in this group.

## 3. Results

The analysis included 936 cases and 5616 controls. Table 1 shows the characteristics of the children in the study. Regarding distribution by histological subtype, the most relevant contingent were ALL cases, with 758 subjects (81%). The ALL proportion was slightly higher in boys than in girls. AML was the next most important group, with 144 cases (15.4%). 

Regarding the sensitivity analysis, for the first analysis we were able to identify more than half of the cases (515; 55%) with the same address at birth and diagnosis and less than half younger than 3 years old (410; 43.8%). Compared to the main group of cases, for the <3-years-old sensitivity group we found a difference between ALL and AML distribution. PM_10_ levels at place of residence and surrounding greenness values were also slightly different.

Table 2 and Figure 2 show the ORs for total leukaemia, ALL, AML subtypes, and both sensitivity groups’ blue-space exposure. Estimations from total leukaemia models show decreased ORs for all distances to blue spaces. Looking at individual results, the lowest and statistically significant one was that associated with residence at <250 m (OR = 0.77; 95%CI = 0.60–0.97). Both total estimated leukaemia model and the sub-models (ALL and sensitivity groups) show an increasing trend from the 500 m exposure buffer: as the distance to blue space grows, the ORs seem to rise. Comparing these groups to the AML group, this presents the opposite trend: we see a reduced OR associated with residence at 1000 m (OR = 0.70; 95%CI = 0.49–0.99).

## 4. Discussion

In this study, we explored the possible association between urban and natural blue spaces and childhood leukaemia incidence in different Spanish regions, taking into account several exposure distances. Our findings showed a reduced OR of childhood leukaemia in zones close to blue spaces. To our knowledge, this study is the first to explore this relationship; consequently, we should be cautious with its interpretation. 

On the one hand, the oceans are essential to human health and well-being. The benefits to the general population are related to wellbeing, mental health, and physical improvement thanks to being places where people can interact socially or do physical activities, for example [7]. Focusing on children, blue spaces have been associated with the same possible effects [10,11]. 

On the other hand, there are some risks related with water. With climate change and increasing pollution of oceans and rivers as a consequence of human activity, all these factors increase the risk of disease vectors [13,31] and microbes responsible for infectious diseases [32]. In addition, indirect impacts on health related to recreational and occupational activities in these spaces could increase skin cancer and sunburn risk [14,33]. All these hazards can be amplified (rise further) by human activities (i.e., pollutants and chemical products) [34,35]. The importance of atmospheric deposition has been confirmed for polycyclic aromatic hydrocarbons (PAHs) and pesticides in the Mediterranean Sea, where atmospheric inputs were higher than riverine ones [36,37]. PAHs are proven human carcinogens and have been linked to multiple human cancers. These depositions are dominated by low-molecular-weight PAHs [38]. 

Blue-space studies on their possible effects agree only that there is no well-established methodology for how these spaces can be studied. Some researchers have only analysed coastal and/or salt-water blue spaces [39,40] while others have combined fresh and salt-water [41,42]. This could distort the results of coastal or inland residents’ possible interactions with blue spaces. Blue-space exposure has also been explored using varying methods. Among these, percentage of blue space around home residence or blue-space prevalence in a buffer are the most used, while questionnaires asking about the time spent in blue spaces are the least common [7]. 

Regarding spatial measurement tools, the most used are land-use maps. Land-use maps have been used in many spatial studies to explore green spaces and crops [21,43]. Their minimum mapping unit ranges from 0.25 ha (Urban Atlas) to 25 ha (CORINE), so they can provide many possibilities to study any land type [44]. However, the characteristics of this study did not allow us to combine both types of land-use database. The most accurate maps did not provide information for all the entire provinces studied while the least accurate maps were not useful for studying urban zones. For that reason, we chose the SIOSE database, whose characteristics allowed us to study urban zones in detail. Furthermore, we took into account possible urban, river course, and coastline changes, which may have altered proximity to blue spaces over the years; to this end, we decided to use three datasets (2005, 2009, and 2011).

Returning to the heterogeneity exposure measurement, it was difficult to establish a buffer size selection properly. Previous blue-space studies have used a great variety of distances, ranging from 100 m to 50 km [42,45]. As yet, there is no available information on how to optimally define minimum distances for urban blue or green space exposure; consequently, we decided to establish four consecutive exposure buffers to explore whether the distance from children’s places of residence to blue spaces could be involved in childhood leukaemia incidence taking into account our previous research [21]. Even though we did not take into consideration the street network or physical barriers (major roads or railways), it appears that the ORs increase as the exposure distance increases for all the models except for AML cases, which follow the opposite trend. Large exposure distances were not evaluated in this study. We wanted to explore the influence of the residential area that could be covered on foot by children or pregnant women from their home residence.

It has been hypothesised that blue spaces may follow similar causal pathways to those identified for urban green spaces including physical activity increment, stress reduction, promotion of social contacts, or reducing extreme temperatures [6]. However, there is no evidence of how blue spaces could modify the previous relationships established between childhood leukaemia and other environmental exposures such as industrial installations, road traffic contamination, or pesticides used for crop growth [19,20,46]. Furthermore, some studies indicate that living in urban areas could increase leukaemia risk, but there is no evidence about what role this could take [47]. Finally, we should not forget individual factors such as family history´s or parent´s lifestyles that could also act as confounders [48]. 

Previous studies have introduced socio-economic status as a potential modifier [18,49,50]. In our case, we introduced this ecological variable in our logistic regression models to control for the potential effects. Due to the lack of individualised socio-economic data, we could only use the 2001 census data, which contained all Spanish census-tract-level socioeconomic information. In our case, neither socio-economic status nor activity rate seemed to affect the relationship explored. Other individual factors and children’s interactions with the environment, such as activities carried out or time spent in these spaces, were not possible to include due to the lack of individual data recorded in the databases used.

There are studies that show that blue spaces are a significant part of natural and urban environments and are correlated with outdoor green spaces and their outcomes [7]. We decided to introduce them into our models as confounding variables by applying NDVI data. Despite satellite images having been used to measure surrounding greenness in many studies, their use to study blue spaces is not usual. NDVI limitations are related to their temporal and spatial availability, so some studies try to use only the most favourable acquisitions to extract NDVI or calculate an annual average [51,52] Moreover, NDVI does not differentiate between private and public spaces, so the values for green spaces can be overestimated. For that reason, we also included urban green spaces extracted from the SIOSE database.

Focusing on sensitivity analysis, our first analysis objective was to explore if there were cases whose home address was different between birth and diagnosis time. The change of residence could bias the results; however according to official data in Spain, approximately only 1% of children change their residence to a different province [53]. To reduce the effect of this potential limitation, we identify the cases with the same address in the two moments comparing both registries. We managed to identify 515 cases with the same address at birth and at diagnosis, 55% of cases. Nevertheless, we could not absolutely establish whether the remaining 45% of cases moved between birth and diagnosis—due to recording differences between RETI-SEHOP and the Birth Registry. 

The second sensitivity analysis’s aim was to evaluate the possible effect of these spaces during pregnancy itself. We decided to select those cases younger than 3 years old for this (43.8%). Despite both sensitivity analyses showing similar results to the cases in the main group, the <3-years-old group have a higher overall OR for the 250 m exposure buffer. This could be derived from the histologic subtype distribution, in which AML represented 20% of the cases versus 15% in the total sample. 

Regarding AML, this leukaemia subtype showed the opposite relationship with blue spaces to overall leukaemia. Previous studies, stratifying by leukaemia type, indicate that exposure to benzene and low-aromatic hydrocarbons are most strongly correlated with AML [54]. Although the influences of human activities, even of the tides, on the air–water exchange of PAHs and others carcinogenic compounds are limited [55], they could play an important role in the transport and diffusion of these chemicals in coastal cities, which requires further study. A major impediment to developing estimates of the cancer attributable to ocean pollutants is lack of detailed, population-level studies of human exposure to marine pollutants.

## 5. Conclusions

As far as we are aware, this is the first study exploring the effects of blue spaces on childhood leukaemia incidence. Our results could point to a possible link between blue-space proximity and this disease. International cooperative transdisciplinary research is needed to identify and weigh up the effect of the prevention and control of blue-space pollution on cancer risk. Despite this, this study can be taken into account for further research to promote blue spaces as a potential health tool for children, so we would encourage a focus on their potential benefits.

## Figures and Tables

**Figure 1 ijerph-19-05232-f001:**
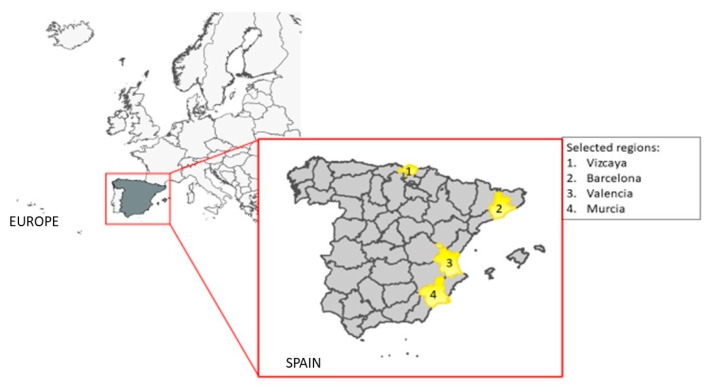
Spanish regions selected for the study.

**Figure 2 ijerph-19-05232-f002:**
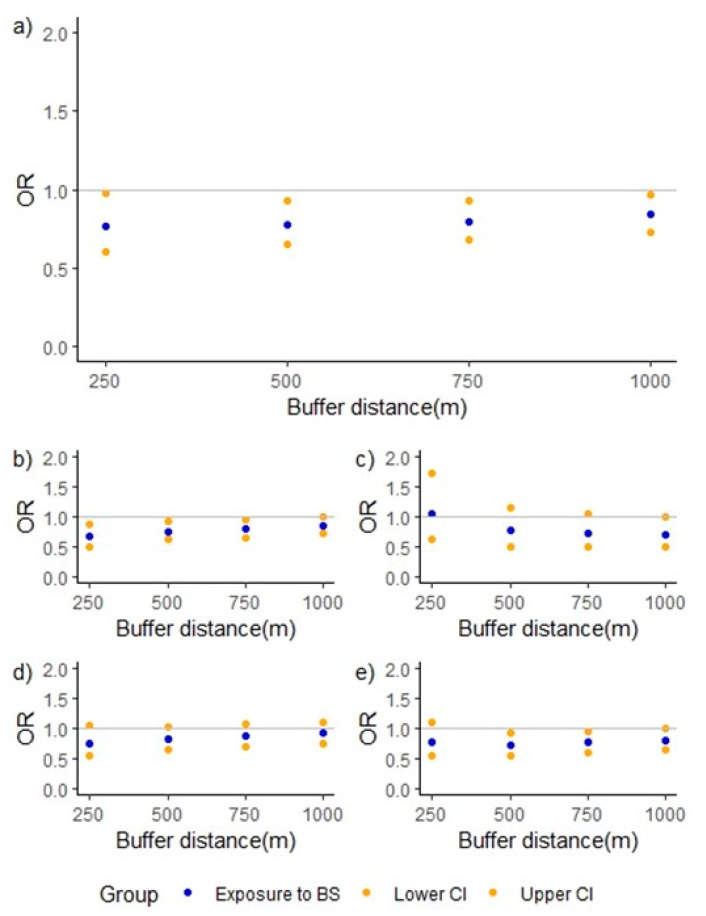
Adjusted models’ graphical representations. Blue points represent ORs, blue-space exposure for each buffer distance compared to the reference group (subjects who did not have any blue space in their 1 km exposure buffers). Lower and upper 95%CIs are represented by orange points. (**a**) Overall childhood leukaemia, (**b**) ALL subtype, (**c**) AML subtype, (**d**) Cases with same address at birth and diagnosis, (**e**) cases <3 years old.

**Table 1 ijerph-19-05232-t001:** Childhood leukaemia cases, sensitivity groups, and controls’ characteristics.

Characteristics	Controls (n = 5616)	Cases (n = 936)	Same Address (n = 515)	*p* Value ^a^	Cases < 3 Years (n = 410)	*p* Value ^a^
Sex, n (%)						
Boy	3126 (55.7%)	521 (55.7%)	283 (55%)		224 (54.6%)	
Girl	2490 (44.3%)	415 (44.3%)	232 (45%)	0.794 ^#^	186 (45.4%)	0.122 ^#^
Age at diagnosis, mean (SD)	x	4.9 (0.12)	5.0 (3.6)	0.625 *	2.0 (1)	x
Activity Rate, mean (SD)	76.3 (4.8)	76 (0.15)	75.8 (4.7)	0.535 *	75.9 (4.5)	0.790 *
SES, mean (SD)	1.06 (0.13)	1.05 (0.12)	1.05 (0.12)	0.605 *	1.04 (0.12)	0.7264 *
Histologic subtype, n (%)						
ALL	x	758 (81.0%)	421 (81.8%)		305 (74.4%)	
AML	x	144 (15.4%)	78 (15.1%)		84 (20.5%)	
CML	x	7 (0.7%)	4 (0.8%)		2 (0.5%)	
Other specific leukaemia	x	17 (1.8%)	7 (1.4%)		13 (3.2%)	
Non-specific leukaemia	x	10 (1.1%)	5 (0.9%)	0.974 ^#^	6 (1.4%)	0.062 ^#^
BSs within 250 m buffer, n (%)						
Yes	708 (12.6%)	102 (10.9%)	55 (10.7%)		46 (11.2%)	
No	4908 (87.4%)	834 (89.1%)	460 (89.3%)	0.898 ^#^	364 (87.8%)	0.032 ^#^
BSs within 500 m buffer, n (%)						
Yes	1630 (29%)	240 (25.6%)	135 (26.2%)		99 (24.2%)	
No	3986 (71%)	696 (74.4%)	380 (73.8%)	0.812 ^#^	311 (75.8%)	0.340 ^#^
BSs within 750 m buffer, n (%)						
Yes	2353 (41.9%)	354 (37.8%)	205 (39.8%)		151 (36.8%)	
No	3363 (58.1%)	582 (62.2%)	310 (60.2%)	0.457 ^#^	259 (63.2%)	0.119 ^#^
BSs within 1000 m buffer, n (%)						
Yes	2934 (52.2%)	455 (48.6%)	264 (51.3%)		196 (52.2%)	
No	2682 (47.8%)	481 (51.4%)	251 (48.7%)	0.334 ^#^	214 (47.8%)	0.074 ^#^
DGUR, n (%)						
Densely populated area	5042 (89.8%)	867 (92.6%)	476 (92.4%)		378 (92.2%)	
Non-densely populated areas	574 (10.2%)	69 (7.4%)	39 (7.6%)	0.889 ^#^	32 (7.8%)	0.781 ^#^
PM_10_ levels at residence, mean (SD)	26.02 (0.07)	26.2 (5.41)	25.9 (5.32)	0.314 *	25.0 (5.29)	<0.01 *
Surrounding greenness (NDVI), median (IQR)						
in 250 m buffer	−0.06 (0.10)	−0.07 (0.11)	−0.06 (0.11)	0.904 ^¥^	−0.05 (0.15)	0.012 ^¥^
in 500 m buffer	−0.04 (0.11)	−0.05 (0.13)	−0.05 (0.13)	0.948 ^¥^	−0.03 (0.16)	0.014 ^¥^
in 750 m buffer	−0.03 (0.12)	−0.03 (0.14)	−0.03 (0.14)	0.898 ^¥^	−0.02 (0.16)	0.013 ^¥^
in 1000 m buffer	−0.02 (0.13)	−0.03 (0.14)	−0.03 (0.15)	0.851 ^¥^	−0.01 (0.17)	0.016 ^¥^

^a^—*p* value from main group of cases compared to sensitivity group. ^#^ Chi-square test for categorical variables, ^¥^ Kruskal–Wallis test, * Student’s *t*-test. Abbreviations: SD—standard deviation, IQR—interquartile range, ALL—acute lymphoblastic leukaemia, AML—acute myeloblastic leukaemia, CML—chronic myeloblastic leukaemia, BSs—blue spaces, DGUR—degree of urbanisation, NDVI—Normalized Difference Vegetation Index.

**Table 2 ijerph-19-05232-t002:** Blue-space analysis for overall childhood leukaemia, ALL and AML subtypes, and identified cases with same address at birth and diagnosis results. Adjusted models for sex, birth year, SES, activity rate, DGUR, PM_10_ levels, and surrounding greenness.

Analysis Group	Exposure Category	Controls (n)	Cases (n)	Adjusted OR (95%CI) ^a^	*p* Value
Childhood Leukaemia					
	Reference	2682	481	-	
	250 m	708	102	0.77 (0.60–0.97)	0.031
	500 m	1630	240	0.78 (0.65–0.93)	0.006
	750 m	2353	354	0.80 (0.68–0.93)	0.005
	1000 m	2934	455	0.84 (0.72–0.97)	0.019
ALL subtype					
	Reference	2682	388	-	
	250 m	708	73	0.68 (0.51–0.89)	0.006
	500 m	1630	188	0.75 (0.62–0.91)	0.004
	750 m	2353	283	0.79 (0.67–0.94)	0.008
	1000 m	2934	370	0.85 (0.72–1.00)	0.049
AML subtype					
	Reference	2682	78	-	
	250 m	708	23	1.06 (0.63–1.71)	0.818
	500 m	1630	39	0.77 (0.50–1.15)	0.210
	750 m	2353	55	0.72 (0.50–1.04)	0.084
	1000 m	2934	66	0.70 (0.49–0.99)	0.045
Same address					
	Reference	2682	251	-	
	250 m	708	55	0.77 (0.56–1.04)	0.100
	500 m	1630	135	0.82 (0.65–1.03)	0.097
	750 m	2353	205	0.87 (0.71–1.07)	0.198
	1000 m	2934	264	0.92 (0.76–1.12)	0.407
Cases < 3 years					
	Reference	2682	214	-	
	250 m	708	46	0.79 (0.55–1.10)	0.172
	500 m	1630	99	0.73 (0.56–0.94)	0.017
	750 m	2353	151	0.77 (0.61–0.97)	0.025
	1000 m	2934	196	0.81 (0.66–1.01)	0.056

^a^—OR and 95% CI for each distance buffer taking as reference subjects without blue space in each buffer. Abbreviations: BS—blue space, OR—odds ratio, CI—confidence interval, ALL—acute lymphoblastic leukaemia, AML—acute myeloblastic leukaemia, SES—socioeconomic status, DGUR—degree of urbanisation.

## Data Availability

Not applicable.

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
