# Peer review of "Exploring Blue Spaces’ Effects on Childhood Leukaemia Incidence: A Population-Based Case–Control Study in Spain"

_ijerph, 2022, doi:10.3390/ijerph19095232_

Round 1
Reviewer 1 Report
This study is quite interesting and novel and has explored the potential relation between blue spaces and childhood leukemia incidence. Data used is reliable and representative. Although the analysis approach used in this MS is relatively robust and traditional, problems are solved. Results of this MS provide important implication that it is necessary to continue studying the potential health effect of blue spaces, especially for the sensitive population.
- All figures and tables should be improved and standardized.
-
Please note the statistical significance in Table 2.
Author Response
Point 1: All figures and tables should be improved and standardized.
Response 1: We have improved and standardized the format of both tables. In addition, we have uploaded all figures and tables in better quality than before in the revised version of the manuscript.
Point 2: Please note the statistical significance in Table 2.
Response 2: We have included the statistical significance in a new column in the revised version of the table 2.

Reviewer 2 Report
A few notes about the text:
1) Table 1: use the acronymous "BSs" defining in the table's caption
2) Table 2: the abbreviation "BS" was not used, remove it from the table's caption
3) lines 273 and 288: correct "stablish" with "establish", as well as "stablished" with "established".
Finally, I think that the method used in the paper can be thoroughly described, by formalizing the model used in the analysis.
Author Response
Point 1: Table 1: use the acronymous "BSs" defining in the table's caption
Response 1: Thank you for point it us. We have use “BSs”
Point 2: Table 2: the abbreviation "BS" was not used, remove it from the table's caption.
Response 2: The abbreviation was removed from the table’s caption.
Point 3: lines 273 and 288: correct "stablish" with "establish", as well as "stablished" with "established".
Response 3: We have corrected them.
Point 4: Finally, I think that the method used in the paper can be thoroughly described, by formalizing the model used in the analysis.
Response 4: The method used in this paper is based on our previous study which explore the relationship between urban green spaces and childhood leukemia (please, see Carlos Ojeda et al., 2021), as we say in the text (‘2.3.2 Environmental covariates’ subsection, third paragraph). We tried to explain briefly it and use this methodology to study blue spaces effect.
- Ojeda Sanchez et al. 2021. Urban green spaces and childhood leukemia incidence: A population-based case-control study in Madrid. Environ Res 2021 Nov;202:111723. doi: 10.1016/j.envres.2021.111723. Epub 2021 Jul 19.

Reviewer 3 Report
Introduction
The introduction of article is too general and requires a broader description of the possible impact of living near polluted blue spaces on the development of diseases in adults and children. Some information can be transfer on this topic from the discussion section to the introduction section. I am thinking of the content from row 231 to 261.
Method
The Authors correctly selected statistical method for their research. There is no description of the 4 regions from which the samples of children were taken to study for develop the research problem. We do not know is these regions are more industrial, agricultural, tourist areas, with clean or polluted air. Why were these regions selected for research. These information could support the description and explanation of the results obtained in the Authors' study.
Discussion
The discussion includes theoretical issues that should be transferred to the introductory part. It is about the content from lines 231 - 261. The research results indicate a strong correlation of green spaces on the level of leukemia incidence, but it is omitted in the discussion. If this value has such a significant impact on the incidence rate, it may be worth considering this fact in the title of the article - „Exploring blue and green spaces’ effects on childhood leukemia….
Research indicates that close distance to the blue spaces (250 m) reduces the number of ALL leukemias and increases the number of AML leukemia in children. The Authors do not try to explain this relationship / regularity. Do children with AML live closer to more industrialized blue spaces areas than children with ALL?
The distance types to the blue spaces adopted by the Authors are small. Every day we travel much longer distances, using for example public transport, a private car or walking. Throughout the day, we breathe air in our city, which circulates in different directions – distance it doesn’t matter. It seems that the adoption of the distance to blue spaces variable as a determinant of childhood leukemia development is unjustified. This relationship seems to be incidental (p values 0.03), while the obtained data indicate a strong negative relationship with the occurrence of leukemia in children and the proximity to green spaces (p values 0.01 for all 4 distances).
In the limitations of the research, the Authors should mention that logistic regression does not examine the influence of one variable on another, but rather conditions it. Generally, Authors should avoid the word “influence” when selecting such statistical methods.
The Authors should explain the criteria for adopting distances from water bodies (250 meters, 500 meters, 750 meters). Is the quality of air and land different at these distances?
Has it been controlled?
The Authors did not provide an analysis of water quality (and this parameter may be important in the research, since we are studying the importance of blue spaces for the development of the disease). Perhaps it is not the distance from water but its quality that determine the development of leukemia in children.
The Authors should explain why exposure to green spaces in the city helps to reduce the incidence of leukemia in children, and in the case of blue spaces the opposite is true. What can this be the result of?
Author Response
Comment 1: The introduction of article is too general and requires a broader description of the possible impact of living near polluted blue spaces on the development of diseases in adults and children. Some information can be transfer on this topic from the discussion section to the introduction section. I am thinking of the content from row 231 to 261.
Response 1: Thank you for your recommendation. According to the reviewer’s suggestion, we have transferred part of the Discussion section to the Introduction section, as the referee can see in the revised version of the manuscript.
Comment 2: The Authors correctly selected statistical method for their research. There is no description of the 4 regions from which the samples of children were taken to study for develop the research problem. We do not know is these regions are more industrial, agricultural, tourist areas, with clean or polluted air. Why were these regions selected for research. These information could support the description and explanation of the results obtained in the Authors' study.
Response 2: Due to the low incidence rate of childhood leukemia and in order to maximize the statistical power of our analysis having cases in the exposure areas, we chose 4 regions with large urban cores and populated areas close to blue spaces.
Comment 3: The discussion includes theoretical issues that should be transferred to the introductory part. It is about the content from lines 231 - 261. The research results indicate a strong correlation of green spaces on the level of leukemia incidence, but it is omitted in the discussion. If this value has such a significant impact on the incidence rate, it may be worth considering this fact in the title of the article - „Exploring blue and green spaces’ effects on childhood leukemia….
Response 3: Thank you for your point of view. We have transferred part of the Discussion section to the Introduction section (see response to Comment #1).
On the other hand, the aim of this paper was to explore the effect of residential proximity to blue spaces on childhood leukemia risk, taking into account the methodology used in our previous paper about proximity to green spaces and risk of childhood leukemia. In the present paper, exposure (proximity) to green spaces was included in the models as a potential confounder, as we say in the Materials and methods section, Statistical analysis subsection.
Comment 4: Research indicates that close distance to the blue spaces (250 m) reduces the number of ALL leukemias and increases the number of AML leukemia in children. The Authors do not try to explain this relationship / regularity. Do children with AML live closer to more industrialized blue spaces areas than children with ALL?
Response 4: As you point out, AML and ALL have different relationships with blue spaces. AML is the second subtype most frequent but it´s etiology and incidence in childhood is quite different. We tried to introduced this possible issue at lines 330-338, where we pointed out that exposure to benzene and low aromatic hydrocarbons are most strongly correlated with AML. Although the influences of human activities on the air-water exchange of some carcinogenic compounds are limited, they could play an important role in the transport and diffusion of these substances in coastal cities, which requires further research. Our epidemiologic study has an exploratory nature that could establish some guidelines of investigation and encourage other researches to continue with this line of research about environmental factors and risk of childhood cancer.
Comment 5: The distance types to the blue spaces adopted by the Authors are small. Every day we travel much longer distances, using for example public transport, a private car or walking. Throughout the day, we breathe air in our city, which circulates in different directions – distance it doesn’t matter. It seems that the adoption of the distance to blue spaces variable as a determinant of childhood leukemia development is unjustified. This relationship seems to be incidental (p values 0.03), while the obtained data indicate a strong negative relationship with the occurrence of leukemia in children and the proximity to green spaces (p values 0.01 for all 4 distances).
Response 5: We have focused on small distance because we wanted to explore the influence of the residential area that could be covered on foot by children or pregnant women from their home residence. We think that 1 km is a maximum reasonable walking distance
As yet there is no available information of how to optimally define minimum distances for blue space exposure, we used the same methodology and buffers of distances applied for our group in a previous study for exposure to green spaces.
Lastly, the p-value of 0.032 shown in Table 1 in relation to the ‘BSs within 250m buffer’ characteristic is referred to the difference between the number of childhood leukemia cases in the main group who has (or no) BSs in a 250m buffer and the number of childhood leukemia cases in the sensitivity group of cases <3 years who has (or no) BSs in a 250m buffer. This p-value is not referred to a possible relationship between proximity to BSs and childhood leukemia. On the other hand, the p-values shown in Table 1 in relation to the ‘Surrounding greenness (NDVI)’ characteristic (0.012, 0.014, 0.013, and 0.016) are referred to the differences between the leukemia cases of the main group and the sensitivity group of leukemia cases <3 years, not the occurrence of leukemia in children and proximity to green spaces.
Comment 6: In the limitations of the research, the Authors should mention that logistic regression does not examine the influence of one variable on another, but rather conditions it. Generally, Authors should avoid the word “influence” when selecting such statistical methods.
Response 6: In the manuscript, we have applied the usual methodology in this type of case-control studies, using mixed multiple unconditional logistic regression models to estimate Odds ratios (ORs) (see Garcia-Perez et al., 2019). Several covariates (including sociodemographic and environmental variables) where included in the models as potential confounders, to control these effects in the relationship between proximity to blue spaces and risk of childhood leukemia. We do not use the word ‘influence’ in the text to refer such statistical models.
- Garcia-Perez et al. 2019. Methodological approaches to the study of cancer risk in the vicinity of pollution sources: the experience of a population-based case-control study of childhood cancer. Int J Health Geogr 2019 May 28;18(1):12. doi: 10.1186/s12942-019-0176-x.
Comment 7: The Authors should explain the criteria for adopting distances from water bodies (250 meters, 500 meters, 750 meters). Is the quality of air and land different at these distances? Has it been controlled?
Response 7: Quality of air and land is difficult to control at any distance. Our database of cases covers 20 years in which most of the cities did not have air quality measurement stations. It´s impossible to introduced the possible air quality with a huge precision. We tried to reduce this issue applying data of particulate matter (PM10) available from the European Environment Agency as we have mentioned in the Materials and methods selection.
Comment 8: The Authors did not provide an analysis of water quality (and this parameter may be important in the research, since we are studying the importance of blue spaces for the development of the disease). Perhaps it is not the distance from water but its quality that determine the development of leukemia in children.
Response 8: This study, as we have mentioned in different sections of the paper, is the first exploratory analysis of the relationship between urban blue spaces and childhood leukemia incidence. With the tools and data used for this study is not possible to measure the quality of the water bodies. Further studies are necessary to explore how water quality could be related with childhood leukemia.
Comment 9: The Authors should explain why exposure to green spaces in the city helps to reduce the incidence of leukemia in children, and in the case of blue spaces the opposite is true. What can this be the result of?
Response 9: the main conclusions of the present paper suggest a reduced risk of childhood leukemia in zones close to blue spaces, i.e., a lower childhood leukemia incidence and residential proximity to blue spaces. In relation to the proximity to green spaces, we have used this covariate as a potential confounder, but we do not shown its associated Odds ratio (OR). The findings related to this concept are shown in our previous paper (see Ojeda Sanchez et al., 2021), where we pointed out a possible association between lower incidence of childhood leukemia and proximity to green spaces. Both studies point out the same conclusions: lower incidence of childhood leukemia in zones close to green (Ojeda Sanchez et al., 2021) and blue (present paper) spaces. The only discrepancy lies in the results referred to the AML histological type and proximity to blue spaces, where we have already addressed this point in the discussion section, lines 330-338
- Ojeda Sanchez et al. 2021. Urban green spaces and childhood leukemia incidence: A population-based case-control study in Madrid. Environ Res 2021 Nov;202:111723. doi: 10.1016/j.envres.2021.111723. Epub 2021 Jul 19.

Round 2
Reviewer 3 Report
The reviewer's suggestions were taken into account. Thank all Authors.
Reviewer
This manuscript is a resubmission of an earlier submission. The following is a list of the peer review reports and author responses from that submission.